# dnaHNet: A Scalable and Hierarchical Foundation Model for Genomic Sequence Learning

**Arnav Shah** [* 1 2]  **Junzhe Li** [* 1 2]  **Parsa Idehpour** [3 4]  **Adibvafa Fallahpour** [1 2 3 5]  **Brandon Wang** [6 7]
**Sukjun Hwang** [6 8]  **Bo Wang** [1 2]  **Patrick D. Hsu** [3 9]  **Hani Goodarzi** [3 10]  **Albert Gu** [6 8]

## Abstract

Genomic foundation models have the potential to decode DNA syntax, yet face a fundamental tradeoff in their input representation. Standard fixed-vocabulary tokenizers fragment biologically meaningful motifs such as codons and regulatory elements, while nucleotide-level models preserve biological coherence but incur prohibitive computational costs for long contexts. We introduce dnaHNet, a state-of-the-art tokenizer-free autoregressive model that segments and models genomic sequences end-to-end. Using a differentiable dynamic chunking mechanism, dnaHNet compresses raw nucleotides into latent tokens adaptively, balancing compression with predictive accuracy. Pretrained on prokaryotic genomes, dnaHNet outperforms leading architectures including StripedHyena2 in scaling and efficiency. This recursive chunking yields quadratic FLOP reductions, enabling $> 3\times$ inference speedup over Transformers. On zero-shot tasks, dnaHNet achieves superior performance in predicting protein variant fitness and gene essentiality, while automatically discovering hierarchical biological structures without supervision. These results establish dnaHNet as a scalable, interpretable framework for next-generation genomic modeling.

## 1. Introduction

The genome encodes the fundamental instructions governing cellular function, development, and evolution (Consortium et al., 2001; enc). Deciphering the syntax of DNA remains a central challenge in modern biology with direct implications for disease diagnosis, drug discovery, and synthetic biology (Zhou et al., 2024; Adibi et al., 2025). Foundation models pretrained on large-scale sequence data have emerged as a powerful approach to this challenge, with performance scaling predictably with compute, data, and parameters (Kaplan et al., 2020). Models such as the Nucleotide Transformer (Dalla-Torre et al., 2024), and Evo (Nguyen et al., 2024; Brixi et al., 2026) have been scaled to billions of parameters and trained across diverse organisms, achieving strong performance on variant effect prediction, regulatory element characterization, and sequence generation (King et al., 2025; Fallahpour et al., 2025b).

However, applying foundation models to genomic data presents a fundamental challenge. Unlike natural languages like English, where whitespace provides clear delimiters for tokenization, DNA is a continuous string of nucleotides without explicit boundaries (Ji et al., 2021; Lindsey et al., 2025). Current approaches address this in one of two ways. The first approach relies on fixed tokenization schemes such as k-mers or Byte-Pair Encoding (BPE) applied directly to nucleotide sequences (Zhou et al., 2024; Dalla-Torre et al., 2024; Sennrich et al., 2016). While computationally efficient, these methods impose arbitrary segmentation boundaries that can fragment biologically meaningful units such as codons, transcription factor binding sites, and splice signals (Bostrom & Durrett, 2020; Fallahpour et al., 2025a).

The second approach avoids tokenization entirely by operating at nucleotide-level (single-nucleotide) resolution (Nguyen et al., 2024). This preserves biological coherence but introduces severe computational constraints, as genomic contexts routinely span millions of base pairs (Benegas et al., 2025). The difficulty of scaling attention to such lengths has led to abandoning Transformers in favor of alternative architectures such as Mamba and StripedHyena (Gu & Dao, 2024; Dao & Gu, 2024; Poli et al., 2023; Ku et al., 2025). Despite these efforts, neither approach adequately resolves the tension between computational tractability and biological fidelity.

Given these challenges, a potential solution could be to

---

[*]Equal contribution  [1]University of Toronto [2]Vector Institute [3]Arc Institute [4]University of Pennsylvania [5]University Health Network [6]Cartesia AI [7]Onepot AI [8]Carnegie Mellon University [9]University of California, Berkeley [10]University of California, San Francisco. Correspondence to: Arnav Shah <arnav.shah@mail.utoronto.ca>, Junzhe Li <vicjz.li@mail.utoronto.ca>.

*Proceedings of the 43$^{rd}$ International Conference on Machine Learning*, Seoul, South Korea. PMLR 306, 2026. Copyright 2026 by the author(s).

adapt dynamic tokenization methods, which have been recently introduced for language modeling (Pagnoni et al., 2025; Hwang et al., 2026). H-Net exemplifies this by replacing fixed tokenization with a differentiable chunking mechanism that learns to segment sequences during training, also demonstrating positive preliminary results on genomic modeling (Hwang et al., 2026). This paradigm is appealing for genomics, where biological information is often organized hierarchically and the optimal granularity of representation varies with sequence context (Libbrecht & Noble, 2015). Yet, whether dynamic chunking can discover biologically meaningful segmentation, scale more efficiently than alternative architectures for long genomic contexts, and yield improvements over existing DNA foundation models remains unexplored.

To bridge this gap, we introduce dnaHNet, a tokenizer-free autoregressive model that establishes a new state-of-the-art for genomic foundation models. Built on the H-Net architecture, dnaHNet operates directly on raw nucleotides and learns to compress them into latent chunks through a differentiable routing mechanism (Hwang et al., 2026). The architecture is recursive and hierarchical, allowing multiple stages of compression that naturally mirror the nested organization of genomic information. We train a number of dnaHNet models ranging from 10M to 1B parameters in size on a comprehensive corpus of prokaryotic genomes from the Genome Taxonomy Database to rigorously study its scaling behavior and performance in downstream tasks(Parks et al., 2025).

Our key contributions are as follows:

- Through extensive scaling law analyses, we demonstrate that dnaHNet achieves superior efficiency compared to StripedHyena2 (Ku et al., 2025), the architecture underlying Evo 2 (Brixi et al., 2026). Recursive compression yields quadratic reductions in FLOP cost for the main network, achieving over three times faster inference than Transformer baselines.

- We identify optimal training regimes and architectural modifications necessary for stable hierarchical learning on genomic data, including compression ratio scheduling, encoder-decoder balancing, and initialization strategies.

- We achieve state-of-the-art zero-shot performance on protein variant effect prediction using experimental fitness data from MaveDB (Esposito et al., 2019; Rubin et al., 2025) and on gene essentiality classification via in silico perturbations (Brixi et al., 2026).

- We show that dnaHNet learns biologically meaningful, context-dependent tokenization that adapts to functional regions like codons, promoters, and intergenic regions.

## 2. Related Work

### 2.1. Genomic Foundation Models

The success of large-scale pretraining in natural language processing has inspired foundation models for genomic sequences. DNABERT introduced bidirectional pretraining using k-mer tokenization (Ji et al., 2021), and DNABERT-2 improved efficiency by adopting Byte-Pair Encoding (Zhou et al., 2024). The Nucleotide Transformer scaled this paradigm to 2.5 billion parameters (Dalla-Torre et al., 2024). Recognizing that genomic function depends on long-range interactions, Enformer demonstrated that extended context improves prediction of gene expression and chromatin states (Avsec et al., 2021). HyenaDNA achieved single-nucleotide resolution at context lengths of one million base pairs by replacing attention with subquadratic operators (Nguyen et al., 2023). More recently, Evo introduced a 7 billion parameter model for prediction and generative design across molecular and genome scales (Nguyen et al., 2024), and Evo 2 extended this to all domains of life using the Striped-Hyena2 architecture (Brixi et al., 2026). Domain-specific models such as megaDNA have also emerged for bacteriophage genome analysis (Shao & Yan, 2024). Despite these advances, all existing approaches either rely on fixed tokenization schemes or operate at nucleotide-level resolution with substantial computational overhead.

### 2.2. Tokenization in Sequence Models

Unlike natural language where whitespace provides word boundaries, DNA is a continuous string without explicit delimiters, making tokenization a fundamental challenge (Ji et al., 2021; Lindsey et al., 2025). Early approaches adopted k-mer tokenization with fixed-length substrings. Byte-Pair Encoding (Sennrich et al., 2016) and SentencePiece (Kudo & Richardson, 2018) were subsequently imported from NLP, though these subword methods may be suboptimal for biological sequences (Bostrom & Durrett, 2020). Recent studies confirm that tokenizer choice creates significant performance differences across genomic tasks (Lindsey et al., 2025). Several approaches have attempted biologically informed designs, including hybrid strategies combining k-mer resolution with BPE compression (Sapkota & Rahman, 2025) and context-aware tokenization that preserves reading frames in coding regions (Fallahpour et al., 2025a). Nevertheless, these remain fixed schemes that cannot adapt segmentation granularity based on sequence context.

### 2.3. Long-Range Sequence Architectures

Modeling long-range dependencies is essential for genomics, where regulatory elements influence gene expression across thousands of base pairs. Standard Transformer attention scales quadratically with sequence length, making

it impractical for million-base contexts. Structured State Space models emerged as an alternative, with S4 demonstrating efficient modeling over tens of thousands of steps (Gu et al., 2022) and Mamba further refining this approach (Gu & Dao, 2024). The Hyena operator uses learned long convolutions to achieve subquadratic scaling (Poli et al., 2023), which was further developed into the hybrid Striped-Hyena architecture (Nguyen et al., 2024). However, these advances address computational challenges without resolving the tension between fixed tokenization and biological fidelity.

## 2.4. Hierarchical and Dynamic Tokenization

Recent work has begun replacing fixed tokenization with learned segmentation. The Byte Latent Transformer introduced a tokenizer-free architecture that groups raw bytes into dynamic patches based on local entropy (Pagnoni et al., 2025). H-Net extended this to end-to-end hierarchical modeling with recursive application for capturing multiple levels of abstraction (Hwang et al., 2026). Parallel developments have explored dynamic tokenization for genomics specifically. MergeDNA applies context-dependent token merging to DNA sequences (Li et al., 2026), PatchDNA proposes biologically informed patching strategies (Del Vecchio et al., 2025), and MxDNA introduces an adaptive tokenization mechanism that learns a variable-length vocabulary during training (Qiao et al., 2024). However, these approaches still operate within a tokenization-then-modeling paradigm, where segmentation is learned as a preprocessing step that produces discrete tokens for a downstream model. Whether dynamic tokenization can learn biologically meaningful segmentation jointly with representations and yield improvements over state-of-the-art DNA foundation models has remained unexplored. Our work addresses this gap by applying hierarchical dynamic chunking to genomic sequences at scale.

## 3. dnaHNet

We introduce dnaHNet, a scalable, tokenizer-free foundation model that establishes a new state-of-the-art for genomic sequence learning. By learning tokenization dynamically, dnaHNet overcomes both the biological fidelity limitations of subword tokenization and the computational costs of byte-level modeling. Section 3.1 formalizes the autoregressive modeling objective and details the hierarchical architecture components. Section 3.2 describes architectural modifications specifically optimized for genomic stability. Finally, Section 3.3 outlines the training objective and inference protocols.

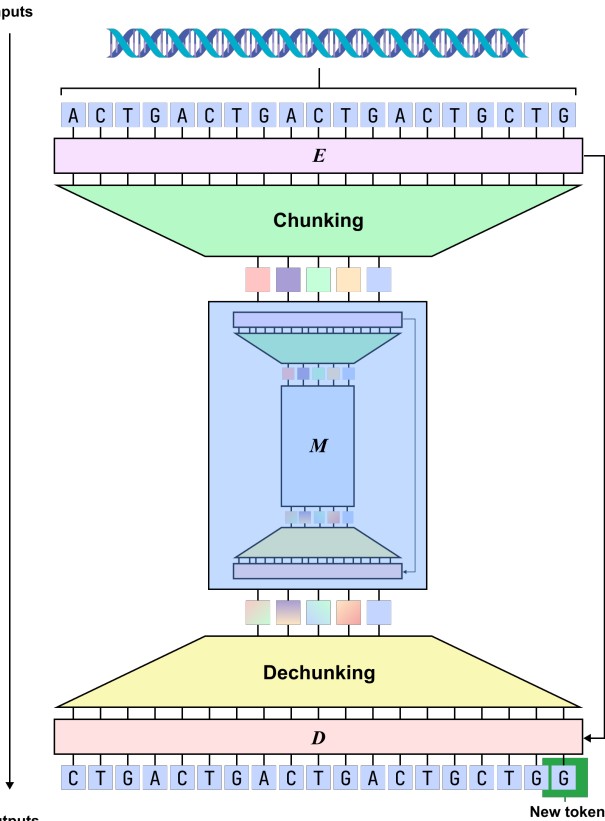

*Figure 1.* **dnaHNet Architecture**. Raw nucleotide sequences are processed by the Encoder (E), which learns segmentation boundaries via a differentiable chunking mechanism. The compressed latent sequence is modeled by the Main Network (M), then upsampled by the Decoder (D), which outputs the next-nucleotide distribution at each position. The left-shifted output relative to the input reflects the standard autoregressive prediction setup (each position predicts the following nucleotide) rather than any explicit copying or preservation of the input by the decoder. The architecture can be applied recursively for multi-level compression.

### 3.1. Architecture

We formulate genomic learning as an autoregressive sequence modeling problem. Given a sequence $X = (x_1, \ldots, x_L)$ where $x_t \in \{A, C, G, T\}$, the goal is to model the probability distribution $P(X) = \prod_{t=1}^{L} P(x_t | x_{<t})$. To achieve this efficiently over long contexts, dnaHNet employs a recursive hierarchy where each module contains three differentiable components: an Encoder ($\mathcal{E}$) that compresses nucleotide-level inputs into latent chunks, a Main Network ($\mathcal{M}$) that processes these latents, and a Decoder ($\mathcal{D}$) that upsamples representations back to nucleotide resolution (Hwang et al., 2026).

**Encoder and Chunking.** The Encoder determines segmentation boundaries through a routing module that identifies information-dense transitions. We employ a hybrid backbone with four Mamba layers (Gu & Dao, 2024) and one

Transformer layer (Vaswani et al., 2017) using sliding window attention (Hwang et al., 2026), keeping the encoder cost linear in input length. The Decoder uses the same backbone. The Encoder transforms input embeddings into hidden states $\mathbf{h}_{1:L} \in \mathbb{R}^{L \times D}$, where $D$ is the model dimension. These states feed into the boundary prediction module which computes boundary probabilities $p_t \in [0, 1]$ via

$$p_t = \frac{1}{2} \left(1 - \text{CosineSim}(W_q \mathbf{h}_t, W_k \mathbf{h}_{t-1})\right) \qquad (1)$$

where $W_q, W_k \in \mathbb{R}^{D \times D}$ are learnable projection matrices. Consecutive nucleotides with dissimilar representations yield high boundary probabilities, encouraging segmentation at contextual shifts such as codon boundaries or regulatory elements. The Chunking layer then downsamples the Encoder output by selecting representations at predicted boundaries $b_t = \mathbb{I}(p_t > 0.5)$, producing a compressed sequence $E = (\mathbf{e}_1, \ldots, \mathbf{e}_{L'})$ where $\mathbf{e}_i \in \mathbb{R}^D$ and $L' \leq L$ (Hwang et al., 2026). To propagate gradients through this discrete selection, we apply a straight-through estimator (Bengio et al., 2013): $b_t$ is treated as a hard decision in the forward pass, while gradients flow backward through the continuous probability $p_t$.

**Hierarchical Sequence Modeling.** The compressed sequence $E$ is processed by the Main Network $\mathcal{M}$, which can be a standard Transformer or another H-Net module, enabling recursive chunking for capturing multiple levels of abstraction. The Main Network produces processed latent states $\hat{E} = (\hat{\mathbf{e}}_1, \ldots, \hat{\mathbf{e}}_{L'}) \in \mathbb{R}^{L' \times D}$. We find that in a two-stage hierarchy, the first stage captures high-frequency local patterns such as codon periodicity, while the second stage models longer-range dependencies across functional regions.

**Decoder and Generation.** The Decoder maps the Main Network outputs $\hat{E}$ back to nucleotide resolution in two steps. First, a smoothing module refines the sequence of latent states $\hat{E} \in \mathbb{R}^{L' \times D}$ into smoothed representations $\bar{E} = (\bar{\mathbf{e}}_1, \ldots, \bar{\mathbf{e}}_{L'}) \in \mathbb{R}^{L' \times D}$ via a recurrence that interpolates discrete chunks:

$$\bar{\mathbf{e}}_j = P_j \hat{\mathbf{e}}_j + (1 - P_j)\bar{\mathbf{e}}_{j-1} \qquad (2)$$

where $j$ indexes the compressed sequence and $P_j$ is the boundary probability associated with the $j$-th chunk. Second, an upsampler expands these smoothed latents to the original sequence length $L$ by copying the vector $\bar{\mathbf{e}}_{c(t)}$ to every nucleotide position $t$ corresponding to chunk index $c(t)$. The Decoder then applies four Mamba layers and one Transformer layer to the upsampled sequence $\tilde{X} \in \mathbb{R}^{L \times D}$ to model autoregressive dependencies. A linear head projects the output to the nucleotide vocabulary logits in $\mathbb{R}^4$, producing the next-nucleotide distribution $P(x_{t+1}|x_{1:t})$.

## 3.2. Improved Techniques for Hierarchical DNA Modeling

Applying hierarchical dynamic chunking to genomic sequences introduces challenges not present in natural language. Through systematic ablations, we identified several modifications critical for stable training and strong downstream performance.

**Model Scale and Capacity Allocation.** We trained models ranging from 10M to 1B total parameters to characterize scaling behavior. A key design decision is the allocation of parameters between the encoder-decoder pair and the main network $\mathcal{M}$. Unlike text, where local byte patterns are relatively simple and the original HNet allocated only 15% of parameters to the encoder-decoder, genomic sequences exhibit complex local dependencies (e.g., codon structure, splice signals) that require substantially greater encoder capacity. We therefore allocate approximately 30% of total parameters to the encoder-decoder—doubling the relative share compared to text HNets—with the remaining 70% allocated to the main network. This balance ensures that the encoder learns sufficiently informative chunk representations to capture genomic structure while the decoder accurately reconstructs nucleotide-level predictions.

**Training Data and Compute-Optimal Recipes.** We varied the total number of pretraining tokens from 5B to over 200B to determine compute-optimal data-to-parameter ratios. Because compression changes the effective token stream length processed by $\mathcal{M}$, standard scaling laws (Kaplan et al., 2020) do not directly apply. We found that optimal dnaHNet configurations prefer training on substantially more data than would be predicted by Chinchilla-style scaling laws (Hoffmann et al., 2022) applied to the raw nucleotide count. Specifically, at a budget of $8 \times 10^{19}$ FLOPs, the optimal dnaHNet model trains on 140B tokens, compared to StripedHyena2 which uses 68B nucleotides and a proportionally larger model (Section 4.3) (Ku et al., 2025).

**Target Compression Ratios.** We set stage-wise target compression ratios based on biological priors. For two-stage models, we use $R_1 = 3$ for the first stage to align with the triplet codon structure of coding regions (Koonin & Novozhilov, 2008). For the second stage, we use $R_2 = 2$, motivated by the phenomenon of codon pair bias, where adjacent codons exhibit non-random co-occurrence patterns that influence translation efficiency and accuracy (Gutman & Hatfield, 1989). This yields an effective compression ratio of $R_1 \times R_2 = 6$, reducing the FLOPs spent by the innermost main network by a factor of $36\times$. We empirically validated this choice with a sweep across (2,2), (3,2), and (3,3) configurations spanning $4\times$ to $9\times$ compression (Section A.4): both (2,2) and (3,2) remain competitive on VEP,

while (3,3) lags noticeably, indicating that $9\times$ compression discards predictive information on prokaryotic sequences. On DEG, the codon-aligned (3,2) outperforms (2,2) across all compute scales, supporting the biological prior empirically.

**Hierarchical Depth Selection.** We explored hierarchies ranging from one to four recursive stages. While deeper hierarchies provide greater compression, they introduce optimization challenges and diminishing returns beyond two stages for our context lengths. Based on scaling experiments (Section 4.3), we adopt a two-stage architecture as the default configuration, achieving $6\times$ compression while maintaining stable training dynamics.

**Auxiliary Loss Weight.** The compression rate regularization coefficient $\alpha$ controls the trade-off between predictive accuracy and adherence to target compression ratios. We found that jointly learning segmentation boundaries and sequence modeling from random initialization can lead to degenerate solutions where the boundary predictor either selects all positions or none. We additionally found that standard compression rate regularization coefficients for natural language ($\alpha \geq 0.03$) were overly aggressive for next nucleotide prediction, necessitating a comparatively smaller value ($\alpha \approx 0.01$) to maintain faithfulness to the target compression ratio.

### 3.3. Training and Inference

**Training Dataset.** We pretrain all models on a processed subset of the Genome Taxonomy Database (GTDB) (Parks et al., 2025), following the filtering, quality control, and dereplication methodology from the OpenGenome dataset by Evo (Nguyen et al., 2024). Genomes are filtered based on assembly completeness, contamination, and marker gene content, retaining a single representative per species-level cluster. The final dataset comprises 17,648,721 sequences totaling 144B nucleotides from 85,205 prokaryotic organisms, with each sequence containing up to 8192 nucleotides extracted as non-overlapping chunks from longer genomes.

**Training Objective.** dnaHNet is trained end-to-end using a composite objective. The primary signal is the autoregressive next-token prediction loss over the nucleotide vocabulary $\mathcal{V} = \{A, C, G, T\}$:

$$\mathcal{L}_{\text{NLL}} = -\sum_{t=1}^{L} \log P_\theta(x_t | x_{<t}) \quad (3)$$

Minimizing $\mathcal{L}_{\text{NLL}}$ alone leads to degenerate segmentation. To regularize the dynamic chunking, we employ the ratio loss from H-Net (Hwang et al., 2026), which guides the model toward a target downsampling ratio $R_s$ for each stage $s$:

$$\mathcal{L}_{\text{rate}}^{(s)} = \frac{R_s}{R_s - 1} \left( (R_s - 1) F_s G_s + (1 - F_s)(1 - G_s) \right) \quad (4)$$

where $F_s = \frac{1}{L} \sum_{t=1}^{L} b_t^{(s)}$ represents the actual fraction of selected chunks (based on discrete decisions $b_t$) and $G_s = \frac{1}{L} \sum_{t=1}^{L} p_t^{(s)}$ is the average boundary probability. This objective aligns the discrete selections with the continuous probability estimates while targeting a compression factor of $R_s$. The total loss is $\mathcal{L} = \mathcal{L}_{\text{NLL}} + \alpha \sum_s \mathcal{L}_{\text{rate}}^{(s)}$.

**Inference.** During inference, boundary probabilities are discretized using threshold $\tau = 0.5$ as $b_t = \mathbb{I}(p_t > \tau)$. When $b_t = 0$, the current nucleotide is accumulated into the encoder state. When $b_t = 1$, the accumulated chunk is dispatched to $\mathcal{M}$, processed, and passed to $\mathcal{D}$, which outputs the next-nucleotide distribution. Sampling proceeds as per normal (e.g. multinomial, greedy decoding, etc.), enabling generation of arbitrary-length sequences with context-dependent granularity.

## 4. Experiments

### 4.1. Evaluation Datasets

We evaluate dnaHNet on three datasets spanning local coding fitness, genome-wide essentiality, and hierarchical structure discovery.

**Protein Variant Effects (MaveDB).** We compiled all 12 nucleotide-level experimental fitness datasets for *E. coli* K-12 from MaveDB (Esposito et al., 2019; Rubin et al., 2025). This dataset of 21250 data points tests the model's ability to capture local coding syntax and predict protein fitness landscapes.

**Gene Essentiality (DEG).** We generated binary essentiality labels for all 62 bacterial organisms in the Database of Essential Genes (DEG) (Luo et al., 2021), with base sequences and annotations from NCBI. Genes matching DEG entries by name or sequence identity ($>99\%$) were labeled essential. This dataset of 185226 data points evaluates the model's capacity to integrate broader genomic context and long-range dependencies.

**Genomic Structure Interpretation (NCBI).** For interpretability analysis, we sourced the *B. subtilis* genome and functional annotations from NCBI (NCBI Resource Coordinators, 2024). We partitioned the genome into distinct functional regions based on annotations to analyze how the model's segmentation aligns with biological structures.

### 4.2. Models and Baselines

We compare dnaHNet against two leading long-sequence architectures. StripedHyena2 (Ku et al., 2025) is the convolu-

tional multi-hybrid architecture underlying Evo 2, which interleaves three different implicit convolution layers with self-attention to improve efficiency over Transformer variants and long convolution architectures such as Hyena or Mamba (Gu & Dao, 2024). We also compare against an optimized Transformer++ architecture (Nguyen et al., 2024) designed for long-context genomic sequences. Additionally, we compare against $k$-mer (with $k \in \{3, 6\}$) and BPE-tokenized Transformer++ variants trained on the same GTDB corpus, with full results in Section A.5. We do not include bidirectional masked language models such as DNABERT-2 (Zhou et al., 2024) and the Nucleotide Transformer (Dalla-Torre et al., 2024), as our zero-shot tasks require comparing autoregressive likelihoods, which are not well-defined under masked objectives.

### 4.3. Scaling Analysis

To assess whether dnaHNet exhibits favorable scaling properties, we conducted scaling law analyses following established methodology (Kaplan et al., 2020; Hoffmann et al., 2022). We trained over 100 models spanning 10M to 1B parameters across three architecture families: dnaHNet (with 1-stage, 2-stage, and 3-stage hierarchies), StripedHyena2, and decoder-only Transformers. For each architecture, we swept model size and training tokens under fixed compute budgets ranging from $4 \times 10^{18}$ to $8 \times 10^{19}$ FLOPs.

To ensure fair comparison, we carefully account for FLOPs across architectures. For dnaHNet, we compute total FLOPs as the sum of encoder, main network, and decoder contributions: $\text{FLOPs}_{\text{total}} = \text{FLOPs}_{\text{enc}}(L) + \text{FLOPs}_{\text{main}}(L/R) + \text{FLOPs}_{\text{dec}}(L)$ where $L$ is the input sequence length and $R$ is the effective compression ratio. Critically, the quadratic attention cost in the main network scales as $\Theta((L/R)^2)$ rather than $\Theta(L^2)$, yielding substantial savings for compressed representations.

**Computational Efficiency.** Figure 2 (left) presents total inference FLOPs as a function of sequence length for compute optimal dnaHNet and StripedHyena2 configurations at the $8 \times 10^{19}$ scale. At $10^6$ nucleotides, dnaHNet (218M) requires $3.89 \times$ fewer FLOPs than SH2 (166M). The right panel of Figure 2 isolates FLOPs per token, revealing the source of dnaHNet's efficiency. While StripedHyena2 exhibits near-linear scaling due to its subquadratic operators, dnaHNet achieves even lower per-token costs through hierarchical compression within the million nucleotide regime. Importantly, the 2-stage dnaHNet (218M) achieves even more efficient performance than its smaller 1-stage variant while containing substantially more parameters in its main network. These theoretical FLOP advantages translate into practical wall-clock gains across throughput, memory, and latency, with the gap widening at longer sequences (Section 4.7).

**Perplexity Scaling.** Figure 3 plots evaluation perplexity against training FLOPs for optimally-configured models of each architecture. We fit power laws of the form PPL $= A \cdot C^{-\alpha}$ where $C$ denotes compute. dnaHNet achieves $\alpha = 0.06$ compared to $\alpha = 0.04$ for StripedHyena2 and $\alpha = 0.01$ for Transformer baselines, indicating demonstrably better compute efficiency.

**dnaHNet exhibits superior scaling efficiency.** Across the compute range tested, dnaHNet consistently achieves lower perplexity than both StripedHyena2 and Transformer baselines at matched FLOPs. The gap widens with increasing compute: at $8 \times 10^{18}$ FLOPs, dnaHNet outperforms StripedHyena2 by 0.078 perplexity points, while at $6 \times 10^{19}$ FLOPs, this gap increases to 0.118 points (Figure 3). Under the optimal scaling regime, StripedHyena2 would require $3 \times 10^{20}$ FLOPs to achieve the same evaluation perplexity as dnaHNet at $8 \times 10^{19}$ FLOPs, representing $3.75 \times$ less efficiency.

**Optimal data-to-parameter ratios differ from standard scaling laws.** Following the Chinchilla methodology (Hoffmann et al., 2022), we computed optimal model sizes and training token counts for each compute budget. Interestingly, dnaHNet's optimal configurations favor training on substantially more tokens than predicted by scaling laws calibrated on non-hierarchical architectures. At $8 \times 10^{19}$ FLOPs, the optimal dnaHNet trains on 140B tokens compared to 68B for StripedHyena2 without plateauing.

### 4.4. Zero-Shot Protein Variant Effect Prediction

We assessed zero-shot performance on the MaveDB dataset, hypothesizing that a model capturing prokaryotic genome statistics would assign lower likelihoods to deleterious variants compared to wild-type sequences. For each gene, we constructed variant sequences by introducing specified mutations and computed autoregressive log-likelihoods for both wild-type and variant sequences. The predicted fitness score was defined as the log-likelihood difference (Figure 4A), with performance quantified via Spearman correlation against experimental fitness.

dnaHNet demonstrated superior predictive accuracy compared to StripedHyena2 when compute-matched, and exhibited stronger scaling with compute budget (Figure 4B). This performance advantage suggests that dynamic chunking effectively compresses and attends to coding regions, enabling finer-grained resolution of fitness landscapes than fixed-scale or nucleotide-level approaches.

### 4.5. Zero-Shot Gene Essentiality Prediction

We evaluated whole-genome modeling by predicting gene essentiality on the DEG dataset. For each gene, we extracted

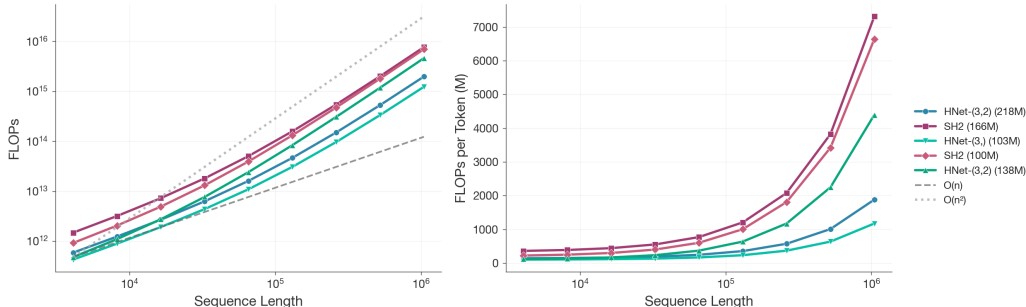

*Figure 2.* **Inference FLOPs.** **(Left)** Total inference FLOPs versus sequence length. At $10^6$ nucleotides, dnaHNet ($218M$) requires $3.89\times$ fewer FLOPs than StripedHyena2 ($166M$). **(Right)** FLOPs per token across sequence lengths. Hierarchical compression enables dnaHNet to achieve lower per-token costs than both linear-scaling baselines and theoretical $O(n)$ and $O(n^2)$ references.

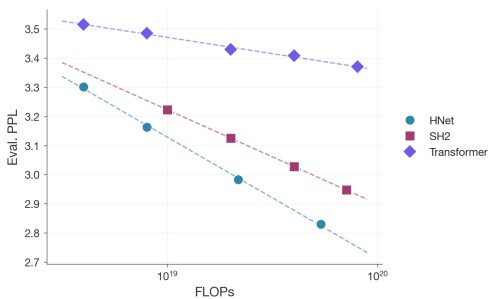

*Figure 3.* Evaluation perplexity scaling. Evaluation perplexity versus training FLOPs for compute-optimal configurations. dnaHNet achieves a scaling exponent of $\alpha = 0.06$ compared to $\alpha = 0.04$ for StripedHyena2 and $\alpha = 0.01$ for Transformers, demonstrating superior compute efficiency across the tested range.

an 8192-bp genomic window centered on the gene to capture local context. Knockout variants were generated by inserting a 15-bp stop codon sequence 12 bp downstream of the start codon (15 nucleotides were replaced). The log-likelihood difference between wild-type and knockout sequences served as the predictor (Figure 5A).

Evaluating performance via AUROC, dnaHNet outperformed StripedHyena2 across all compute budgets, with gains increasing monotonically with compute budget (Figure 5B). This indicates that the hierarchical architecture effectively integrates local coding syntax disrupted by the stop codon with the broader genomic context required to determine gene essentiality.

### 4.6. Biological Structure Hierarchy

A key advantage of dnaHNet is interpretability through its learned chunking boundaries. We analyzed the *B. subtilis* genome by processing five random 49152-bp windows with a trained (3,2) two-stage model and computing token selection rates within each functional region and codon position (Figure 6). Our analysis reveals that dnaHNet learns biological structure in a hierarchical manner (Table 1).

*Table 1.* **Hierarchical Chunking Statistics.** Boundary selection rates across genomic regions, codon positions, and regulatory motifs for two-stage dnaHNet. Stage 1 exhibits triplet periodicity; Stage 2 distinguishes functional boundaries and aligns with regulatory motifs.

| Genomic Feature | Stage 1 | Stage 2 |
|---|---|---|
| *Global Selection Rate* | 35.8% | 51.3% |
| **Functional Regions** | | |
| Promoter | 35.6% | 71.5% |
| Start Codon | 35.1% | 81.3% |
| Coding Region | 35.8% | 48.4% |
| Stop Codon | 34.5% | 51.7% |
| Intergenic | 35.3% | 74.6% |
| **Codon Periodicity** (Coding Region Only) | | |
| Position 1 | 6.5% | 51.8% |
| Position 2 | 42.6% | 38.0% |
| Position 3 | 58.4% | 55.9% |
| **Regulatory Motifs** | | |
| SD region ($-20$ to $-5$) | 35.2% | 77.4% |
| SD motif (exact match) | 34.8% | 84.5% |
| $-10$ box (TATAAT) | 37.7% | 75.0% |
| $-35$ box (TTGACA) | 33.2% | 53.9% |

**Stage 1 (Codon Awareness).** The first stage learns triplet codon structure inherent to coding sequences. While aggregate selection rates across functional regions remain indistinguishable from baseline, coding regions exhibit strong periodicity aligned with codon positions (Table 1). The first position is rarely selected (6.5%), while second and third positions show elevated rates (42.6% and 58.4% respectively).

**Stage 2 (Functional Awareness).** The second stage shifts from local syntax to broader genomic organization. Selection rates diverge significantly across functional regions, with promoters (71.5%), start codons (81.3%), and intergenic regions (74.6%) selected far above coding regions (48.4%). This pattern sharpens at the motif level: Stage 2

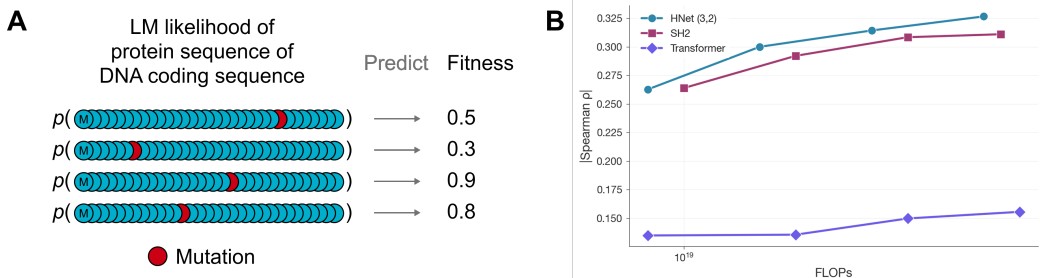

*Figure 4.* **Protein VEP Results. (A)** Schematic of the zero-shot scoring method, using language model likelihood of mutated coding sequences to predict experimental fitness. **(B)** Absolute Spearman correlation on MaveDB benchmarks versus training FLOPs. dnaHNet consistently achieves higher correlation than StripedHyena2 (SH2) and Transformer baselines across all compute budgets.

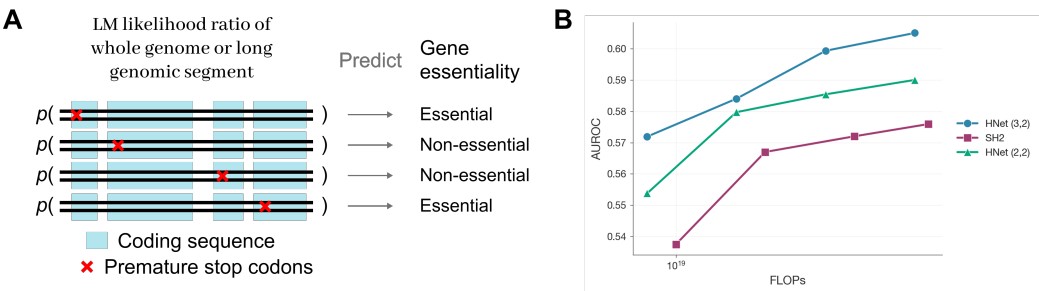

*Figure 5.* **Gene Essentiality Prediction. (A)** Schematic of the *in silico* perturbation task. Gene essentiality is predicted by comparing wild-type likelihood against a variant with inserted premature stop codons. **(B)** Classification AUROC on DEG versus training FLOPs. Both dnaHNet configurations outperform StripedHyena2, with the (3,2) hierarchy demonstrating strongest scaling, hypothesized to be due to matching the underlying biological structure of codons in the first layer.

boundaries align with exact Shine-Dalgarno motif matches (84.5%), broader SD regions (77.4%), and promoter $-10$ boxes (75.0%), while the more degenerate $-35$ box shows only marginal alignment (53.9%). Stage 1 shows no comparable motif preference, consistent with its role in capturing codon-level structure. Together, these results indicate that upper layers leverage the codon-aware tokens constructed in the first stage to recognize functional regulatory elements from raw sequence data without supervision.

### 4.7. Wall-Clock Efficiency

To confirm that dnaHNet's theoretical FLOP advantages translate into practical gains, we benchmarked wall-clock forward-pass performance against StripedHyena2 on a single NVIDIA H100 GPU with BF16 precision, sweeping sequence lengths from $2^{10}$ to $2^{19}$ nucleotides for three dnaHNet variants (M, L, XL) and three StripedHyena2 configurations (100M, 167M, 234M parameters).

dnaHNet achieves substantially better throughput, memory, and latency than size-comparable StripedHyena2 models across all tested sequence lengths, with the gap widening at longer contexts as hierarchical compression reduces the effective length processed by the main network. At $2^{16}$ nucleotides ($\approx$ 65K nt), dnaHNet-XL processes 939K tokens/sec compared to 156K tokens/sec for StripedHyena2-

234M ($6.0\times$). At $2^{18}$ nucleotides ($\approx$ 262K nt), dnaHNet-XL uses 9.7 GB of peak GPU memory and completes a forward pass in 410 ms, compared to 34.3 GB and 2,352 ms for StripedHyena2-234M ($3.5\times$ less memory, $5.7\times$ lower latency). At $2^{19}$ nucleotides, both StripedHyena2-167M and StripedHyena2-234M exceed the 80GB memory budget and fail to run, while all three dnaHNet variants complete the forward pass within memory. Figure 6 shows the full sweep.

## 5. Discussion

We introduced dnaHNet, a tokenizer-free foundation model that achieves state-of-the-art genomic sequence learning. dnaHNet demonstrates superior scaling efficiency compared to StripedHyena2 ($\alpha = 0.06$ versus $\alpha = 0.04$), with recursive compression enabling over $3\times$ more efficient inference at million-nucleotide contexts. On downstream tasks, dnaHNet achieves state-of-the-art zero-shot performance on protein variant effect prediction and gene essentiality classification across all compute scales.

Most notably, dnaHNet learns biologically meaningful segmentation without supervision. The first stage discovers triplet codon structure, while the second stage shifts to functional organization, preferentially marking promoters and intergenic regions. This emergent hierarchy mirrors the

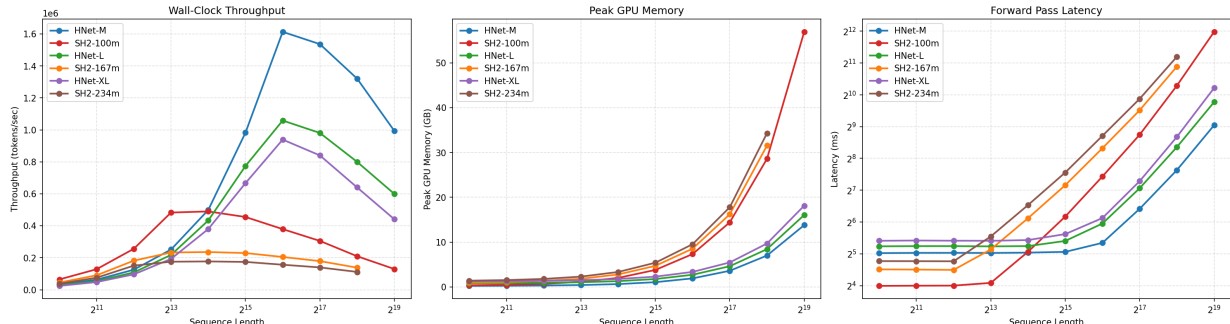

*Figure 6.* **Wall-clock efficiency comparison between dnaHNet and StripedHyena2.** Three dnaHNet variants (M, L, XL) compared against three StripedHyena2 configurations (100M, 167M, 234M parameters) across sequence lengths $2^{10}$ to $2^{19}$ nucleotides on a single NVIDIA H100 GPU. **(Left)** Throughput in tokens per second; dnaHNet variants achieve 3–6× higher throughput than size-comparable StripedHyena2 models. **(Middle)** Peak GPU memory; dnaHNet uses 3–4× less memory at long contexts, with StripedHyena2-167M and StripedHyena2-234M exceeding 80GB at $2^{19}$ nucleotides while all dnaHNet variants fit. **(Right)** Forward-pass latency (log scale); dnaHNet's advantage grows with sequence length as compression shrinks the effective length processed by the main network.

nested organization of genomic information and validates that end-to-end learning can recover biological syntax from raw sequences.

The scaling analysis reveals particularly important implications for compute-efficient genomic modeling. The widening performance gap between dnaHNet and StripedHyena2 as compute increases suggests that hierarchical compression provides compounding benefits at scale. Notably, achieving equivalent perplexity would require StripedHyena2 to expend 3.75× more compute, a substantial efficiency margin that grows with model scale. We also observe that dnaHNet's optimal training regime deviates from standard Chinchilla scaling laws: the architecture benefits from substantially more training tokens relative to parameter count (140B versus 68B tokens at matched compute), likely because compression reduces the effective sequence length processed by the main network, allowing it to extract more signal per raw nucleotide. These findings suggest that as genomic foundation models scale toward trillion-parameter regimes, hierarchical architectures may offer critical efficiency advantages over fixed-tokenization or byte-level alternatives.

### 5.1. Limitations

Several limitations merit discussion. We pretrained exclusively on prokaryotic genomes, which lack the complex regulatory architecture of eukaryotes including introns and long-range chromatin interactions. Our evaluation focused on zero-shot tasks, so behavior under fine-tuning remains unexplored. The fixed target compression ratios, while biologically motivated, may be suboptimal for non-coding or eukaryotic sequences. Finally, our scaling analyses extend only to 1B parameters, leaving large-scale behavior undetermined.

### 5.2. Future Directions

Promising directions include extending pretraining to eukaryotic genomes to test whether dynamic chunking discovers structures such as splice sites and enhancers. The interpretability of learned boundaries suggests applications in discovering novel functional elements. Finally, integrating dnaHNet with protein language models could enable unified biological modeling from genotype to phenotype.

## 6. Conclusion

Existing genomic models face a persistent tradeoff: fixed tokenizers achieve computational efficiency but fragment biological motifs, while nucleotide-level models preserve biological coherence but scale poorly to long contexts. dnaHNet resolves this tradeoff through end-to-end learned segmentation, achieving both the efficiency gains of compressed representations and the biological fidelity of nucleotide-resolution input. The result is a framework that is both a powerful predictive tool and a window into the statistical organization of life. As we scale to larger hierarchies and more diverse taxonomic data, such models may not only predict biological function but assist in designing it, from optimizing synthetic operons to engineering novel protein pathways.

## Impact Statement

This paper presents work whose goal is to advance the field of genomic foundation models. Potential positive impacts include improved understanding of gene function and accelerated biological discovery. As with all genomic modeling tools, dual-use concerns exist, though our focus on prokaryotic genomes and interpretive downstream tasks limits immediate biosecurity risks.

## Acknowledgements

We thank Jesse Lee for their assistance in developing the graphics and figures used in this paper. We are grateful to Joshua Achiam for helpful discussions and suggestions throughout the duration of the project. This work was supported by the University of Toronto, Vector Institute, and Arc Institute.

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

# A. Appendix

## A.1. Model Architecture Details

We experimented with three primary configurations of dnaHNet, Medium (M), Large (L), and Extra Large (XL), to study scaling behavior. All models utilize a two-stage recursive hierarchy. The architectural layout for all configurations follows the pattern:

$$["m4", ["T1m4", ["TN"], "m4T1"], "m4"]$$

where `m4` denotes 4 Mamba layers (Encoder/Decoder blocks), `T1` denotes 1 Transformer layer, and `TN` denotes $N$ Transformer layers in the innermost Main Network ($\mathcal{M}$).

Table 2 details the specific hyperparameters for each model variant used in our scaling laws and downstream evaluations.

*Table 2.* **dnaHNet Model Configurations.** Hyperparameters for the three primary model scales. Dimensions and heads are listed for the Outer / Middle / Inner hierarchy stages respectively.

| Hyperparameter | dnaHNet-M | dnaHNet-L | dnaHNet-XL |
|---|---|---|---|
| **Architecture Layout** | 2-Stage | 2-Stage | 2-Stage |
| **Inner Layers** ($N$) | 7 | 12 | 18 |
| **Model Dimensions** ($D$) | 512 / 640 / 768 | 576 / 704 / 768 | 640 / 768 / 832 |
| **Attention Heads** | 8 / 10 / 12 | 9 / 11 / 12 | 11 / 12 / 13 |
| **Intermediate Dim** | 0 / 1024 / 2048 | 0 / 1024 / 2048 | 0 / 1108 / 2218 |
| **SSM State Dim** | 128 | 128 | 128 |
| **SSM Conv Dim** | 4 | 4 | 4 |
| **Context Window** | 8192 | 8192 | 8192 |

## A.2. Training Hyperparameters

All models were trained on the Genome Taxonomy Database (GTDB) using the hyperparameters listed below. We utilized the AdamW optimizer with a linear warmup and cosine decay schedule.

**Optimization and Stability.** To ensure stable training across the hierarchy, we employed layer-wise learning rate multipliers. The scripts indicate a multiplier schedule of `2.0 1.5 1.0`, applying higher learning rates to the outer compressive layers to encourage rapid convergence of the tokenization boundaries.

Table 3 summarizes the global training settings, and Table 4 provides the specific configuration for the reported training runs.

*Table 3.* **Global Training Hyperparameters.**

| Parameter | Value |
|---|---|
| **Optimizer** | AdamW |
| **Weight Decay** | 0.05 |
| **Gradient Clipping** | 1.0 |
| **Precision** | BF16 Mixed |
| **LR Schedule** | Linear Warmup + Cosine Decay |
| **LR Multipliers** | 2.0 (Outer), 1.5 (Mid), 1.0 (Inner) |
| **Sequence Packing** | Padded |
| **Auxiliary Loss Weight** ($\alpha$) | $0.01 - 0.05$ |

## A.3. Computational Resources

Training was performed on compute nodes equipped with NVIDIA GPUs (A100/H100 class).

- **dnaHNet-M:** Trained on 4 GPUs with 32 CPUs per task.

- **dnaHNet-L:** Trained on 4 GPUs with 32 CPUs per task.

*Table 4.* **Run-Specific Training Settings.** The dnaHNet-XL configuration uses the biologically motivated compression targets ($3 \times 2$) described in the main text.

|  | dnaHNet-M | dnaHNet-L | dnaHNet-XL |
|---|---|---|---|
| Base Learning Rate | $8 \times 10^{-4}$ | $7.5 \times 10^{-4}$ | $6.5 \times 10^{-4}$ |
| Batch Size (per device) | 16 | 8 | 8 |
| Grad Accumulation | 8 | 16 | 16 |
| Target Compression ($R_1, R_2$) | 2, 2 | 3, 2 | 3, 2 |
| Aux Loss Weight | 0.03 | 0.02 | 0.02 |
| Warmup Steps | 4,739 | 4,683 | 6,756 |
| Max Steps | 94,784 | 93,663 | 135,073 |

- **dnaHNet-XL:** Trained on 4 GPUs with 32 CPUs per task (extended duration).

The implementation leveraged Triton and TorchInductor for kernel optimization, with DeepSpeed Stage 2 for memory efficiency.

### A.4. Compression Ratio Sweep

We empirically validated the choice of stage-wise compression targets by training two-stage dnaHNet at $(R_1, R_2) \in \{(2, 2), (3, 2), (3, 3)\}$, spanning $4\times$ to $9\times$ effective compression with all other settings held fixed. Figure 7 shows VEP performance against training FLOPs. (2,2) and (3,2) track each other closely, with (3,2) edging ahead at the highest compute budgets, while (3,3) lags consistently, indicating that $9\times$ compression discards predictive information. We adopt (3,2) as the default, as it also aligns Stage 1 with the triplet codon structure (Table 1) and outperforms (2,2) on DEG (Table 7).

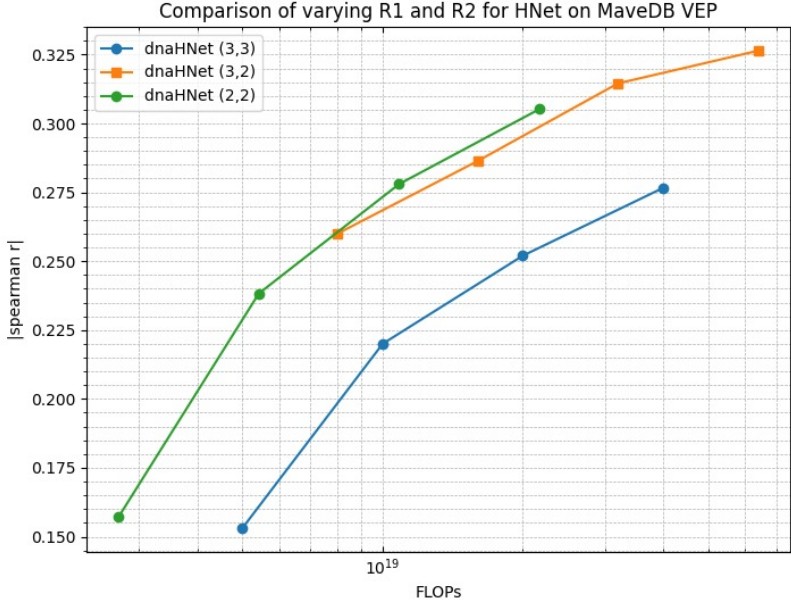

*Figure 7.* **VEP scaling across compression ratios.** Zero-shot Spearman correlation on MaveDB vs. training FLOPs for $4\times$, $6\times$, and $9\times$ effective compression.

### A.5. Additional Baselines

To isolate the contribution of dynamic chunking, we trained tokenized Transformer++ baselines on the same GTDB corpus at matched compute, varying only the tokenization scheme: 3-mer, 6-mer, and a BPE vocabulary learned from GTDB. Figure 8 shows that dnaHNet (3,2) outperforms all three across the full compute range.

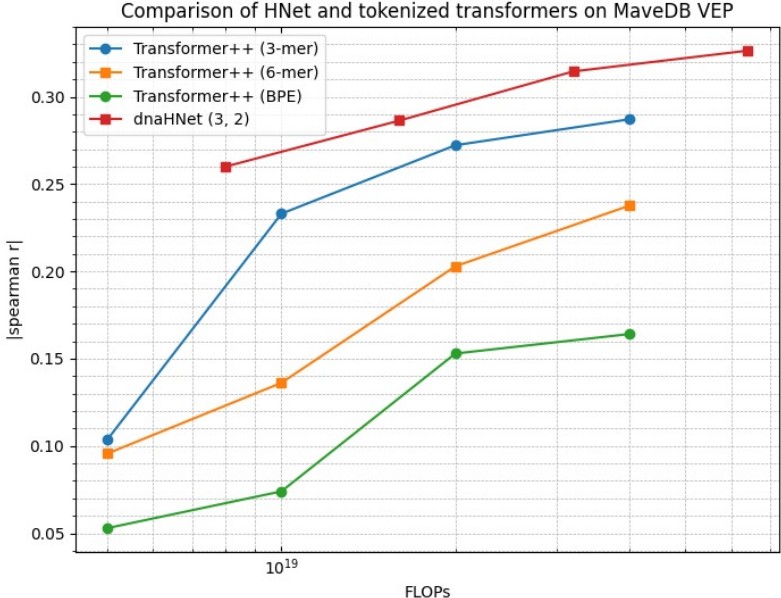

*Figure 8.* **Zero-shot VEP performance vs. tokenized Transformer++ baselines.** Spearman correlation on MaveDB across training FLOPs for dnaHNet (3,2), 3-mer Transformer++, 6-mer Transformer++, and BPE Transformer++.

## A.6. Detailed Downstream Results

We provide the exact numerical results corresponding to the scaling analyses in the main text. Table 5 details the zero-shot Spearman correlations for Protein Variant Effect Prediction (VEP) on MaveDB. Table 6 details the AUROC scores for Gene Essentiality (GE) prediction on the DEG dataset. The exact 15bp stop codon sequence used for the gene essentiality evaluations was the sequence "TAATAATAATAGTGA".

*Table 5.* **Protein Variant Effect Prediction (VEP) Results.** Zero-shot Spearman correlation on MaveDB across compute scales. dnaHNet consistently achieves higher correlation than baselines at comparable FLOP budgets.

| Model | Training FLOPs | Spearman $\rho$ |
|---|---|---|
| **dnaHNet** | $8.00 \times 10^{18}$ | 0.2601 |
| | $1.60 \times 10^{19}$ | 0.2865 |
| | $3.20 \times 10^{19}$ | 0.3143 |
| | $6.40 \times 10^{19}$ | **0.3266** |
| **StripedHyena2** | $1.00 \times 10^{19}$ | 0.2639 |
| | $2.00 \times 10^{19}$ | 0.2921 |
| | $4.00 \times 10^{19}$ | 0.3084 |
| | $7.11 \times 10^{19}$ | 0.3110 |
| **Transformer** | $8.00 \times 10^{18}$ | 0.1348 |
| | $2.00 \times 10^{19}$ | 0.1355 |
| | $4.00 \times 10^{19}$ | 0.1497 |
| | $8.00 \times 10^{19}$ | 0.1555 |

## A.7. Confound Analyses for Gene Essentiality Prediction

To rule out the possibility that the DEG results in Section 4.3 reflect generic sequence-disruption sensitivity or species-familiarity bias rather than functional understanding, we ran three controls. (i) *Synonymous-normalized AUROC*: we subtract the model's log-likelihood drop on a matched neutral perturbation (15 nucleotides replaced with synonymous codons) from the stop-codon-insertion drop, isolating signal specific to premature termination. (ii) *Pooled z-AUROC*: gene scores are

*Table 6.* **Gene Essentiality (GE) Prediction Results.** Classification AUROC on the DEG dataset. The (3,2) hierarchy configuration of dnaHNet demonstrates the strongest scaling behavior.

| Model | Training FLOPs | AUROC |
|---|---|---|
| **dnaHNet (3,2) Hierarchy** | $8.00 \times 10^{18}$ | 0.5719 |
| | $1.60 \times 10^{19}$ | 0.5840 |
| | $3.20 \times 10^{19}$ | 0.5993 |
| | $6.40 \times 10^{19}$ | **0.6050** |
| **dnaHNet (2,2) Hierarchy** | $8.00 \times 10^{18}$ | 0.5538 |
| | $1.60 \times 10^{19}$ | 0.5797 |
| | $3.20 \times 10^{19}$ | 0.5854 |
| | $6.40 \times 10^{19}$ | 0.5900 |
| **StripedHyena2** | $1.00 \times 10^{19}$ | 0.5375 |
| | $2.00 \times 10^{19}$ | 0.5670 |
| | $4.00 \times 10^{19}$ | 0.5720 |
| | $7.10 \times 10^{19}$ | 0.5759 |

standardized within each genome before pooling, removing between-species score-scale differences. (iii) *Length and GC stratification*: AUROC is computed within within-species tertile bins of gene length and GC content. Results are reported in Tables 7 and 8, with full per-genome AUROCs in Table 9. Performance remains well above chance under all three controls, declines only modestly under synonymous normalization (e.g., $0.599 \rightarrow 0.587$ for the (3,2) configuration at $6.4 \times 10^{19}$ FLOPs), and shows no single length or GC regime dominating the signal.

*Table 7.* **Synonymous-normalized and pooled z-scored AUROCs across compute scales.** Macro AUROC is the unweighted mean across 62 DEG genomes ($\pm$ SD). Synonymous-normalized AUROC subtracts the model's response to a matched neutral synonymous-codon perturbation. Pooled z-AUROC standardizes scores within each genome before pooling, removing species-familiarity bias.

| Training FLOPs | Macro (raw) | Macro (syn-norm.) | Pooled z (raw) | Pooled z (norm.) |
|---|---|---|---|---|
| *dnaHNet (2,2) configuration* | | | | |
| $8.00 \times 10^{18}$ | $0.519 \pm 0.029$ | $0.512 \pm 0.017$ | 0.546 | 0.523 |
| $1.60 \times 10^{19}$ | $0.566 \pm 0.067$ | $0.559 \pm 0.059$ | 0.581 | 0.563 |
| $3.20 \times 10^{19}$ | $0.579 \pm 0.058$ | $0.566 \pm 0.043$ | 0.598 | 0.575 |
| $6.40 \times 10^{19}$ | $0.585 \pm 0.067$ | $0.573 \pm 0.054$ | 0.601 | 0.582 |
| *dnaHNet (3,2) configuration* | | | | |
| $8.00 \times 10^{18}$ | $0.574 \pm 0.069$ | $0.552 \pm 0.036$ | 0.587 | 0.566 |
| $1.60 \times 10^{19}$ | $0.589 \pm 0.057$ | $0.574 \pm 0.049$ | 0.603 | 0.580 |
| $3.20 \times 10^{19}$ | $0.599 \pm 0.069$ | $0.582 \pm 0.067$ | 0.600 | 0.579 |
| $6.40 \times 10^{19}$ | $\mathbf{0.599 \pm 0.077}$ | $\mathbf{0.587 \pm 0.068}$ | **0.604** | **0.585** |

*Table 8.* **AUROC stratification by gene length and GC content.** Length and GC tertiles are computed within each genome to control for species-specific distributions; we then report the mean across the 62 DEG genomes ($\pm$ SD). Within-genome spread is the per-genome maximum minus minimum AUROC across its three tertile bins, averaged over genomes.

| Configuration | Overall | Length spread[†] | GC spread[†] |
|---|---|---|---|
| (2,2) | $0.585 \pm 0.068$ | $0.093 \pm 0.061$ | $0.078 \pm 0.047$ |
| (3,2) | $0.599 \pm 0.078$ | $0.104 \pm 0.077$ | $0.090 \pm 0.057$ |

| Configuration | Overall | Len. T1 | Len. T2 | Len. T3 | GC T1 | GC T2 | GC T3 |
|---|---|---|---|---|---|---|---|
| (2,2) | $0.585 \pm 0.068$ | $0.618 \pm 0.101$ | $0.558 \pm 0.069$ | $0.586 \pm 0.054$ | $0.583 \pm 0.074$ | $0.575 \pm 0.064$ | $0.572 \pm 0.069$ |
| (3,2) | $0.599 \pm 0.078$ | $0.637 \pm 0.106$ | $0.567 \pm 0.065$ | $0.595 \pm 0.055$ | $0.607 \pm 0.076$ | $0.579 \pm 0.080$ | $0.570 \pm 0.066$ |

[†]Per-genome (max $-$ min) AUROC over three tertile bins; reported value is the mean $\pm$ SD across 62 genomes.

*Table 9.* **Per-genome AUROC on the DEG gene essentiality task.** Results for the (2,2) and (3,2) dnaHNet configurations at $6.4 \times 10^{19}$ training FLOPs, reporting both raw and synonymous-normalized AUROC. Gene counts include the totals used for each genome.

| RefSeq ID | DEG IDs | AUROC (2,2) raw | AUROC (2,2) norm. | AUROC (3,2) raw | AUROC (3,2) norm. | Total | Essential |
|---|---|---|---|---|---|---|---|
| AE014133.2 | DEG1054 | 0.6002 | 0.5706 | 0.6337 | 0.5848 | 1962 | 201 |
| AJ749949.2 | DEG1052 | 0.6412 | 0.6031 | 0.6587 | 0.6180 | 2051 | 453 |
| AM747720.1 | DEG1066 | 0.6032 | 0.6115 | 0.6197 | 0.6034 | 3585 | 345 |
| AM747721.1 | DEG1066 | 0.5138 | 0.5309 | 0.4531 | 0.5357 | 2884 | 33 |
| AM747722.1 | DEG1066 | 0.3029 | 0.3978 | 0.4292 | 0.5066 | 793 | 14 |
| AM747723.1 | DEG1066 | 0.6591 | 0.7096 | 0.9444 | 0.9697 | 101 | 2 |
| CP017054.1 | DEG1060 | 0.5881 | 0.5673 | 0.5650 | 0.5485 | 3923 | 444 |
| CP029332.1 | DEG1059 | 0.6200 | 0.5950 | 0.6454 | 0.5965 | 4880 | 234 |
| HG941718.1 | DEG1048 | 0.6024 | 0.5850 | 0.6164 | 0.6032 | 4981 | 300 |
| NC_000907.1 | DEG1005 | 0.5357 | 0.5387 | 0.5512 | 0.5549 | 1723 | 626 |
| NC_000908.2 | DEG1006 | 0.5811 | 0.5556 | 0.6087 | 0.5519 | 524 | 377 |
| NC_000912.1 | DEG1068 | 0.5373 | 0.5467 | 0.5837 | 0.5779 | 723 | 334 |
| NC_000915.1 | DEG1008 | 0.5250 | 0.5262 | 0.5508 | 0.5184 | 1558 | 302 |
| NC_000962.3 | DEG1010;DEG1025;DEG1027 | 0.5547 | 0.5453 | 0.5700 | 0.5558 | 3906 | 1061 |
| NC_000964.3 | DEG1001 | 0.6819 | 0.6668 | 0.7024 | 0.6657 | 4240 | 269 |
| NC_002163.1 | DEG1031;DEG1049 | 0.5324 | 0.5169 | 0.5272 | 0.5174 | 1579 | 334 |
| NC_002505.1 | DEG1003;DEG1067 | 0.5442 | 0.5356 | 0.5539 | 0.5454 | 2578 | 519 |
| NC_002506.1 | DEG1003;DEG1067 | 0.4874 | 0.4837 | 0.4852 | 0.4907 | 1014 | 139 |
| NC_002516.2 | DEG1030;DEG1036;DEG1051 | 0.5679 | 0.5687 | 0.5761 | 0.5600 | 5573 | 662 |
| NC_002745.2 | DEG1002 | 0.6649 | 0.6260 | 0.6917 | 0.6368 | 2706 | 303 |
| NC_002771.1 | DEG1014 | 0.5468 | 0.4993 | 0.5595 | 0.5222 | 761 | 291 |
| NC_002952.2 | DEG1065 | 0.6522 | 0.6094 | 0.6503 | 0.6071 | 2811 | 298 |
| NC_002953.3 | DEG1064 | 0.6533 | 0.6074 | 0.6597 | 0.6346 | 2686 | 309 |
| NC_003028.3 | DEG1007 | 0.6055 | 0.5884 | 0.5996 | 0.5900 | 2146 | 204 |
| NC_003062.2 | DEG1045 | 0.6097 | 0.5880 | 0.5949 | 0.5938 | 2720 | 283 |
| NC_003098.1 | DEG1007 | 0.5793 | 0.5720 | 0.6018 | 0.5799 | 2025 | 212 |
| NC_003197.2 | DEG1011 | 0.5854 | 0.5770 | 0.6140 | 0.6055 | 4453 | 225 |
| NC_003295.1 | DEG1057 | 0.5946 | 0.5981 | 0.6079 | 0.6061 | 3473 | 373 |
| NC_003296.1 | DEG1057 | 0.4195 | 0.4784 | 0.4362 | 0.5347 | 1676 | 56 |
| NC_003923.1 | DEG1063 | 0.6399 | 0.6141 | 0.6514 | 0.6247 | 2710 | 259 |
| NC_004347.2 | DEG1029 | 0.6583 | 0.6395 | 0.6529 | 0.6290 | 4369 | 408 |
| NC_004631.1 | DEG1016;DEG1033 | 0.6663 | 0.6524 | 0.6841 | 0.6609 | 4611 | 357 |
| NC_004663.1 | DEG1023 | 0.5977 | 0.5924 | 0.6320 | 0.6176 | 4761 | 288 |
| NC_005296.1 | DEG1041 | 0.5901 | 0.5464 | 0.5797 | 0.5501 | 4959 | 528 |
| NC_005966.1 | DEG1013 | 0.5829 | 0.5707 | 0.5794 | 0.5745 | 3262 | 493 |
| NC_006350.1 | DEG1035 | 0.6438 | 0.6229 | 0.6385 | 0.6084 | 3726 | 238 |
| NC_006351.1 | DEG1035 | 0.5793 | 0.5429 | 0.6179 | 0.5992 | 2704 | 49 |
| NC_007432.1 | DEG1042 | 0.6382 | 0.6247 | 0.6231 | 0.6052 | 2084 | 300 |
| NC_007595.1 | DEG1040 | 0.3818 | 0.4273 | 0.3909 | 0.4273 | 56 | 1 |
| NC_007604.1 | DEG1040 | 0.5769 | 0.5642 | 0.5841 | 0.5733 | 2666 | 617 |
| NC_007650.1 | DEG1024 | 0.5616 | 0.5851 | 0.5952 | 0.6145 | 2463 | 41 |
| NC_007651.1 | DEG1024 | 0.6625 | 0.6370 | 0.6737 | 0.6548 | 3396 | 330 |
| NC_007795.1 | DEG1017;DEG1061 | 0.6484 | 0.6076 | 0.6321 | 0.6149 | 2767 | 379 |
| NC_008463.1 | DEG1015 | 0.6243 | 0.6023 | 0.6360 | 0.6242 | 5966 | 345 |
| NC_008601.1 | DEG1012 | 0.5979 | 0.5875 | 0.6197 | 0.5968 | 1791 | 391 |
| NC_008787.1 | DEG1050 | 0.5503 | 0.5379 | 0.5569 | 0.5260 | 1607 | 364 |
| NC_009009.1 | DEG1021 | 0.6353 | 0.6107 | 0.6556 | 0.6206 | 2274 | 209 |
| NC_009085.1 | DEG1043;DEG1044 | 0.6246 | 0.6100 | 0.6065 | 0.5905 | 3351 | 606 |
| NC_009511.1 | DEG1028 | 0.5961 | 0.5668 | 0.5924 | 0.5673 | 4972 | 473 |
| NC_010079.1 | DEG1062 | 0.6360 | 0.6207 | 0.6411 | 0.6059 | 2810 | 295 |
| NC_010729.1 | DEG1022;DEG1039 | 0.6195 | 0.6175 | 0.6285 | 0.6239 | 2031 | 411 |
| NC_011375.1 | DEG1038 | 0.6559 | 0.6128 | 0.6786 | 0.6506 | 1718 | 180 |
| NC_011916.1 | DEG1020 | 0.5816 | 0.5480 | 0.5880 | 0.5578 | 3886 | 441 |
| NC_012925.1 | DEG1058 | 0.5316 | 0.5228 | 0.5374 | 0.5301 | 1889 | 333 |
| NC_014171.1 | DEG1047 | 0.5174 | 0.5153 | 0.5095 | 0.5048 | 5348 | 477 |
| NC_014375.1 | DEG1046 | 0.6253 | 0.5919 | 0.6310 | 0.5924 | 3337 | 377 |
| NC_016776.1 | DEG1034 | 0.5518 | 0.5366 | 0.5841 | 0.5773 | 4323 | 447 |
| NC_016810.1 | DEG1032 | 0.6344 | 0.6264 | 0.6548 | 0.6451 | 4643 | 343 |
| NC_016856.1 | DEG1026 | 0.5686 | 0.5470 | 0.5467 | 0.5430 | 4632 | 105 |
| NC_022240.1 | DEG1055 | 0.6103 | 0.6024 | 0.6211 | 0.6103 | 2286 | 604 |
| NZ_CP008957.1 | DEG1056 | 0.5359 | 0.5236 | 0.5266 | 0.5281 | 5453 | 1067 |
| NZ_LR881938.1 | DEG1018;DEG1019 | 0.5297 | 0.5043 | 0.5246 | 0.5043 | 4340 | 666 |

