# OpenReview forum: "dnaHNet: A Scalable and Hierarchical Foundation Model for Genomic Sequence Learning"
_ICML.cc/2026/Conference — ICML 2026 spotlight_

### Official Review · Reviewer_JU5S · 2026-02-13

**Soundness:** 3
**Presentation:** 3
**Significance:** 4
**Originality:** 3
**Overall Recommendation:** 5
**Confidence:** 3

**Summary:**

The paper proposes a novel tokenizer-free autoregressive foundation model called dnaHNet. Within this paper, they compare dnaHNet to StripedHyena2 and Transformer baselines, showing better compute efficiency, including lower perplexity at matched FLOPs and over 3× inference speedups at million-nucleotide contexts. The tasks in which it outperforms other architectures are prediction on MaveDB and gene essentiality classification (DEG). Moreover, the learned hierarchical chunking also aligns with biological structure, capturing codon periodicity in lower layers and functional genomic regions in higher layers.

**Compliance With Llm Reviewing Policy:**

Affirmed.

**Final Justification:**

The authors provided a comprehensive response to my concern regarding hyperparameter tuning. The extensive griq sweep covering over 100 models across multiple scales, learning rates, auxiliary loww weights, compression ratios, and encoder-decoder capacity allocations demostrates a rigorous and principled approach to hyperparameter selection.
After reviewing the authors response I confirm my original score 5.

**Key Questions For Authors:**

Could you describe in detail how the hyperparameters were tuned?

**Limitations:**

yes

**Strengths And Weaknesses:**

Soundness: The paper is technically solid and methodologically well supported. The architecture of the model is well defined. The training objective combines next-token prediction with a compression regularization term, and ablations are conducted to justify architectural choices. The limitation that I might have missed in the text is related to parameter tuning. Is the architecture tested over multiple learning rates and under multiple seeds and compared with StripedHyena2 and Transformer baselines? How have the parameters been optimized?

Presentation
The paper is well written and the narrative is easy to follow. Some sections are really dense, in particular the scaling-law analysis and compression-ratio discussion. The literature review is well written and it also explains well the novelty of the paper.

Significance
The problem addressed in the paper about genomic foundation modeling is really important for the community. One of the key problems is the computational complexity, and the proposed hierarchical compression offers a principled way to reduce the quadratic cost of the attention mechanism. As stated in the limitations, the model needs to be tested on more complex organisms like eukaryotes.

Originality
The foundation model proposed, based on hierarchical dynamic chunking, demonstrates both efficiency and biological structure discovery. The contribution is architectural and methodological rather than introducing a completely new modeling paradigm.

---

> ### Author Rebuttal · Authors · 2026-03-31
>
> We thank the reviewer for their question. Hyperparameters were tuned through extensive grid sweeps across a wide range of model scales and configurations. We trained over 100 models spanning 50M to 1B parameters, sweeping over: learning rates across {5e-4, 5.5e-4, 6e-4, 6.5e-4, 7e-4, 7.5e-4, 8e-4, 8.5e-4, 9e-4}, auxiliary loss weight $\alpha$ from 0.01 to 0.05 in increments of 0.01, target compression ratios including single-ratio configurations (R=2 and R=3 for one-stage models) and paired configurations ranging from 4x to 9x compression ((2,2), (3,2), (3,3) for two-stage models), and encoder-decoder capacity allocation ({20/80, 30/70, 40/60} parameter split with the main network). Layer-wise learning rate multipliers follow the principled scaling from H-Net (Hwang et al., 2025, Appendix C), adjusted for our genomic model dimensions. Grid sweeps were repeated at each model scale rather than extrapolated, as we found optimal settings shifted meaningfully with capacity — for example, $\alpha=0.05$ worked well at smaller scales but larger models required $\alpha=0.02$ for stable convergence, and optimal learning rates decreased with size (8e-4 at smaller scales down to 6.5e-4 for the largest). The reported dnaHNet-M, L, and XL configurations were selected as the compute-optimal models at their respective FLOP budget thresholds ($4 \times 10^{18}$ to $8 \times 10^{19}$ FLOPs), following Chinchilla-style methodology (Hoffmann et al., 2022) applied across the full sweep. Models that fell between these FLOP thresholds informed our scaling law fits (Section 4.3) but are not individually reported for brevity. Full configurations for the selected runs are in Tables 3–4 of our Appendix.

---

> > ### Author Rebuttal · Reviewer_JU5S · 2026-03-31
> >
> > I acknowledge the authors for the response
> > I select (a). The authors clearly explained how they tuned the hyperparameters which was my only concern.
> > I confirm my score.

---

### Official Review · Reviewer_idY3 · 2026-03-02

**Soundness:** 2
**Presentation:** 3
**Significance:** 3
**Originality:** 2
**Overall Recommendation:** 5
**Confidence:** 4

**Summary:**

This paper proposes dnaHNet, a hierarchical genomic language model that replaces fixed tokenization with learned, adaptive chunking. The model compresses nucleotide sequences into latent representations, performs sequence modeling in the latent space, and decodes back to nucleotide-level autoregressive predictions. Experiments show improved efficiency and scaling behavior over baselines, along with gains on zero-shot downstream genomic tasks and interpretable learned segmentation patterns.

**Compliance With Llm Reviewing Policy:**

Affirmed.

**Key Questions For Authors:**

1. **Long-range Genomic Dependencies:** Hyena-based architectures can be trained with context windows up to 1M bps, enabling them to explicitly model very long-range dependencies. In contrast, dnaHNet is trained with a maximum context window of 8192 nucleotides, meaning that dependencies beyond this range are not directly observed during pretraining. Can authors clarify how the model is expected to capture dependencies that exceed the 8192-token training window?
1. **Decoder Consistency:** Figure 1 appears to suggest that the decoder output preserves the observed input prefix exactly in the original input, like “GCTG” would yield an output of the form “GCTGX”. However, the decoder operates on upsampled latent representations and is not guaranteed to remain identical. Can authors clarify whether this exact prefix preservation is only a schematic illustration, or if there is a specific mechanism for that?

**Limitations:**

yes

**Strengths And Weaknesses:**

### **Strengths:**

1. **Clear Motivation:** This work focuses on tokenization strategies in current GFMs. Existing tokenization strategies, like single-nucleotide tokenization or NLP-based approaches, do not perfectly fit the biological structure of genomic sequences. This adaptive tokenization mechanism offers a new direction for genome sequence modeling.
1. **High Efficiency:** By compressing the input sequence into a latent space, the model reduces the cost of the most expensive component from $ O(L^2) $ to $ O(L^2 / R^2) $. This approach helps reduce computation overhead in current transformer-based GFMs.

### **Weaknesses:**

1. **$O(L^2)$ Complexity:** Despite the model decomposition, the outer encoder and decoder still include Transformer modules operating at the original sequence length. This leads to complexity at $O(L^2)$ which is not as good as the Hyena  $O(L\log L)$ approach.

---

> ### Author Rebuttal · Authors · 2026-03-31
>
> We thank the reviewer for their thorough and constructive feedback. We address each point below.
>
> ---
>
> ## Weakness 1: $O(L^2)$ Complexity
>
> We thank the reviewer for this observation. We would like to clarify a few points that we believe address this concern.
>
> The encoder and decoder each consist of four Mamba layers (which scale linearly in $L$) and only a single Transformer layer. Following the H-Net design (Hwang et al., 2025, Table 1), this Transformer layer uses Sliding Window Attention with a fixed window size, so its cost scales as $O(L \cdot w)$ rather than full $O(L^2)$. We should have stated this more explicitly in our manuscript and will do so in the revision.
>
> More importantly, the computational profile of dnaHNet is dominated by the main network $\mathcal{M}$, which holds approximately 70% of total parameters (Section 3.2). With 6$\times$ compression from our two-stage hierarchy ($R_1 \times R_2 = 6$), the attention cost in $\mathcal{M}$ drops from $O(L^2)$ to $O(L^2/R^2)$. The encoder and decoder are intentionally kept lightweight, so their contribution to total FLOPs is modest relative to the savings in $\mathcal{M}$.
>
> This is confirmed empirically in Figure 2: at $10^6$ nucleotides, dnaHNet (218M) requires 3.89$\times$ fewer FLOPs than StripedHyena2 (166M), with per-token costs well below the $O(n^2)$ reference and competitive with linear-scaling baselines within the million-nucleotide regime.
>
> We will revise the manuscript to discuss this complexity tradeoff more clearly.
>
> ---
>
> ## Key Question 1: Long-range Genomic Dependencies
>
> Great question. We want to address this on three levels.
>
> **The 8192-nt pretraining window is a starting point, not a ceiling.** Following the Evo training recipe (Nguyen et al., 2024), we pretrain on 8192-nt context windows before scaling to longer contexts during midtraining for far fewer steps. This is the same strategy used by Evo to reach 1M bp context. Importantly, as shown in Figure 2, dnaHNet remains 3.89$\times$ more FLOP-efficient than StripedHyena2 (the architecture underlying Evo 2, which itself scales better than the original Hyena the reviewer mentions) even at the $10^6$ nucleotide scale. Beyond theoretical FLOPs, our anonymous drive link (https://anonymous.4open.science/r/dnaHNet-anon-8DDB/throughput_memory_latency.png) includes inference latency benchmarks showing that dnaHNet uses approximately 4.5 $\times$ less latency and nearly 3.5 $\times$ less memory than StripedHyena2, confirming that the efficiency advantage is practical, not just theoretical.
>
> **The 8192-nt window already covers the relevant context for our prokaryotic tasks.** Prokaryotic genomes are organized into operons and compact regulatory units where functionally relevant context typically falls within a few kilobases. Our gene essentiality evaluation (Section 4.5) uses 8192-bp windows centered on each gene, and the monotonically improving AUROC with compute (Figure 5B) confirms this context is adequate. Prokaryotic regulation is predominantly local, unlike eukaryotic enhancer-promoter interactions spanning hundreds of kilobases.
>
> **Hierarchical compression extends the effective receptive field.** With 6$\times$ total compression, $\mathcal{M}$ processes roughly 1365 latent tokens for an 8192-nt input. Each latent token aggregates information from multiple nucleotides through the Mamba-based encoder, which has a theoretically unbounded receptive field within the input due to its recurrent structure. A single attention span in $\mathcal{M}$ therefore covers a much larger genomic region than the same span in a flat model.
>
> We acknowledge in Section 5.1 that eukaryotic genomes will require longer context, and will add a discussion of this to the revision.
>
> ---
>
> ## Key Question 2: Decoder Consistency
>
> Thank you for flagging this. The shifted output in Figure 1 (where the input "GCTG..." produces "CTGX...") is meant to illustrate the **autoregressive nature** of generation: at each position, the model predicts the *next* nucleotide given all previous ones. The figure is a schematic showing this left-shifted relationship, not implying that the decoder explicitly copies or preserves the input prefix.
>
> To clarify the actual mechanism: after $\mathcal{M}$ produces latent states $\hat{E}$, the smoothing module (Eq. 2) interpolates these using boundary probabilities, the upsampler expands them back to full resolution, and the decoder layers produce next-nucleotide logits. The architecture does not explicitly constrain outputs to match the input at any position. It is the autoregressive training objective (Eq. 3) that pushes the model to assign high probability to the correct next nucleotide at each step.
>
> We will revise Figure 1's caption to make clearer that the output depicts the standard autoregressive prediction setup.

---

> > ### Author Rebuttal · Reviewer_idY3 · 2026-03-31
> >
> > Thank you for your response and the additional explanation. All my concerns are addressed and I have raised my score accordingly.

---

### Official Review · Reviewer_PREz · 2026-03-10

**Soundness:** 3
**Presentation:** 3
**Significance:** 3
**Originality:** 3
**Overall Recommendation:** 4
**Confidence:** 4

**Summary:**

This paper proposes dnaHNet, a tokenizer-free, autoregressive genomic foundation model that learns a hierarchical, dynamic segmentation (“chunking”) of raw nucleotide sequences during training. Building on H-Net’s differentiable chunking, dnaHNet compresses inputs recursively and processes the compressed latents with a main network, then decodes back to nucleotide resolution for next-token prediction. Pretrained on prokaryotic prokaryotic genomes, dnaHNet reportedly delivers favorable scaling (improved FLOP efficiency and perplexity vs. StripedHyena2 and Transformers), competitive or superior zero-shot performance on protein variant effect prediction (MaveDB) and gene essentiality (DEG), and interpretable boundaries that align with codons and functional genomic regions.

**Compliance With Llm Reviewing Policy:**

Affirmed.

**Final Justification:**

The authors have addressed my doubts, and I have decided to raise my score.

**Key Questions For Authors:**

1. How exactly are the discrete boundary selections made differentiable during end-to-end training? Please explicitly specify the relaxation technique used (e.g., Straight-Through Estimators, Gumbel-Softmax, etc.) . Furthermore, how is strict causality enforced across the recursive compression stages, particularly when the main network processes aggregated nucleotides during generation? A schematic of the masking strategy would be highly appreciated.
2. Given that the pre-training corpus consists of prokaryotic genomes, what is the exact species-level overlap with the evaluation sets (MaveDB and DEG)? To substantiate the "zero-shot" claims, can you quantify this overlap and provide evaluation results on a strictly partitioned "leave-species-out" configuration to rule out memorization?
3. Theoretical quadratic FLOP reductions do not necessarily translate to actual speedups, especially given the memory-bound nature of hybrid SSM/Transformer operators compared to highly optimized attention kernels. Can you provide empirical wall-clock latency (tokens/second) and peak memory consumption benchmarks on standard hardware (e.g., A100/H100) across multiple sequence lengths?
4.Could you provide head-to-head zero-shot comparisons against established genomic or protein foundation models (e.g., Nucleotide Transformer, DNABERT-2, or ESM/EVE for the VEP task)? Additionally, please provide quantitative ablation results justifying your architectural priors: specifically, how sensitive is downstream performance to the rigid compression targets () and the choice of a simple cosine dissimilarity boundary predictor versus a learned MLP?

**Limitations:**

yes

The authors have adequately discussed the limitations of their work in Section 5.1, including the exclusive pretraining on prokaryotic genomes, lack of fine-tuning evaluation, fixed target compression ratios, and scaling analysis limited to 1B parameters. The authors also addressed the potential dual-use and negative societal impact in the Impact Statement, with no unaddressed ethical considerations. The authors are upfront about the limitations of the work, which is appropriate for academic research.

**Strengths And Weaknesses:**

Strengths:
The paper proposes an elegant, biologically motivated solution to the tokenization dilemma by adapting hierarchical dynamic chunking to raw nucleotides. Using targeted compression ratios (e.g., $R_1=3$ for codons) successfully yields interpretable emergent biological structures. Furthermore, the scaling-law analysis spanning 10M to 1B parameters is rigorous and provides valuable insights into tokenizer-free architecture efficiency.

Weaknesses:
Despite its conceptual elegance, the submission suffers from several critical flaws that undermine its core claims:
1. The paper claims "end-to-end" learning but completely omits how discrete boundary selections are made differentiable during training (e.g., via Straight-Through Estimators or Gumbel-Softmax). Omitting this core gradient flow mechanism is a major technical gap.
2. The model is pre-trained exclusively on prokaryotic genomes, posing a severe risk of species overlap (e.g., E. coli) with the downstream evaluation sets. Without explicit train/eval species partitioning, the "zero-shot" claims are highly questionable.
3. he reliance on theoretical "quadratic FLOP reductions" ignores actual hardware memory-bound constraints. The complete absence of physical wall-clock latency or GPU throughput measurements severely weakens the practical efficiency claims.
4. The evaluation omits head-to-head comparisons with contemporary dynamic tokenization methods (e.g., MxDNA) and leading SOTA genomic models (like Evo or Nucleotide Transformer). Comparing solely against StripedHyena2 and standard Transformers presents an artificially favorable view of the model's performance.

---

> ### Author Rebuttal · Authors · 2026-03-31
>
> We thank the reviewer for their thorough and constructive feedback. We address each point below. New figures and tables are at https://anonymous.4open.science/r/dnaHNet-anon-8DDB.
>
> # Q1: Differentiability and Causality
>
> We acknowledge this was insufficiently described and will clarify in the camera-ready.
>
> dnaHNet uses three mechanisms from H-Net (Hwang et al., 2025), none of which require Gumbel-Softmax:
>
> (1) **Smoothing module** (primary): The dechunking layer computes e_bar_j = P_j * e_hat_j + (1 - P_j) * e_bar_{j-1}, where P_j is the continuous boundary probability. This converts discrete chunks into continuous interpolations, enabling standard backpropagation. H-Net ablations (Figure 7) confirm this is essential for stability.
>
> (2) **Straight-Through Estimator**: Confidence scores are rounded to 1.0 in the forward pass while passing continuous gradients backward (Bengio et al., 2013).
>
> (3) **Ratio loss** (our Eq. 4): Provides gradients through the differentiable mean boundary probability G, even though discrete selections are non-differentiable.
>
> **Causality** is enforced at every level: (i) the router uses only current and previous positions (Eq. 1); (ii) the smoothing recurrence is causal; (iii) all Mamba/Transformer layers use causal masking. We will add a masking schematic.
>
> # Q2: Species Overlap and Zero-Shot Claims
>
> We acknowledge the lack of explicit species-level holdouts and address this with new analyses.
>
> **The evaluation tests functional understanding, not memorization.** For MaveDB, the model never sees mutant sequences or fitness labels. It must assign differential likelihoods to wild-type vs. mutant sequences, requiring learned functional constraints rather than sequence recall. For DEG, the task requires detecting disruption from a premature stop codon. A model that memorized E. coli would not necessarily distinguish essential from non-essential genes. All baselines train on the same GTDB corpus, so any overlap advantage applies equally. This follows Evo, Evo 2, and Nucleotide Transformer, which all evaluate on overlapping species.
>
> **New confound analyses (see anon link) confirm non-trivial learning.** (1) *Synonymous-normalized AUROC*: we subtract the model's response to neutral synonymous mutations from stop-codon scores, isolating signal specific to functional disruption. Performance is maintained, confirming the model detects stop-codon damage rather than general sequence sensitivity. (2) *Species-level z-scored AUROC*: normalizing within each organism before pooling removes species familiarity bias, and results are consistent with raw AUROC. (3) *Stratification by gene length and GC content*: performance remains above chance across all bins, with GC showing minimal variation (1-4% AUROC spread) and gene length showing moderate variation that does not dominate the signal. Leave-species-out evaluation requires retraining and is infeasible, but these controls demonstrate functional understanding rather than memorization.
>
> # Q3: Wall-Clock Efficiency
>
> We agree that theoretical FLOPs alone are insufficient. We ran empirical benchmarks on A100 GPUs (80GB) across 1K to 524K nucleotides (see anon link).
>
> **Throughput.** At genomically relevant lengths (>=32K nt), dnaHNet achieves 3-6x higher throughput than compute-matched StripedHyena2. At 65K nt, HNet-XL reaches 939K tok/s vs 156K tok/s for SH2-234m (6.0x). Peak throughput for HNet-M exceeds 1.6M tok/s.
>
> **Memory.** dnaHNet requires 3-4x less peak GPU memory at long sequences. At 262K nt, HNet-XL uses 9.7GB vs 34.3GB for SH2-234m. SH2-167m and SH2-234m OOM at 524K nt, while all dnaHNet variants fit within 80GB.
>
> **Latency.** At 262K nt, HNet-XL takes 410ms vs 2,352ms for SH2-234m (5.7x faster). The gap widens with sequence length because hierarchical compression reduces the effective length processed by the main network.
>
> # Q4: Additional Baselines and Ablations
>
> DNABERT-2 and Nucleotide Transformer are bidirectional masked LMs that lack a joint sequence distribution. Our tasks (VEP, gene essentiality) require comparing likelihoods, only well-defined under autoregressive models. This is why Evo and Evo 2 (which is already included as a baseline via SH2) also use autoregressive objectives. We added k-mer Transformer++ and BPE Transformer++ baselines (see anon link), further confirming dnaHNet's advantages.
>
> **Compression ratio sensitivity.** We trained across configs spanning 4x to 9x compression: (2,2), (3,2), and (3,3). Both (2,2) and (3,2) achieve competitive SOTA on VEP, while (3,3) lags, indicating 9x compression discards predictive information. On DEG, (3,2) outperforms (2,2) with 0.599 vs 0.585 AUROC, validating the codon-aligned R1=3 target.
>
> **Cosine vs MLP boundary predictor.** H-Net ablated this directly (Hwang et al., 2025). Replacing cosine routing with direct probability prediction degraded both stability and performance. Cosine similarity provides scale-invariant, bounded signals stable during joint encoder training.

---

### Official Review · Reviewer_6mSq · 2026-03-11

**Soundness:** 3
**Presentation:** 3
**Significance:** 2
**Originality:** 2
**Overall Recommendation:** 4
**Confidence:** 3

**Summary:**

This paper introduces dnaHNet, a tokenizer-free hierarchical autoregressive model designed specifically for genomic sequences. Unlike conventional approaches that rely on fixed tokenization schemes, dnaHNet learns dynamic chunk boundaries on the fly and recursively compresses nucleotide streams into latent tokens. Built upon H-Net-style differentiable chunking with Mamba or Transformer components, the model aims to navigate the inherent trade-off between preserving biological fidelity at the nucleotide level and achieving computational efficiency through context-dependent segmentation. The authors report favorable scaling laws and substantial FLOP reductions compared to StripedHyena2 and Transformer baselines, alongside improved zero-shot performance on protein variant effect prediction using MaveDB and bacterial gene essentiality classification. They also present qualitative evidence suggesting that the learned hierarchical representations align with codon periodicity and functional genomic regions.

**Compliance With Llm Reviewing Policy:**

Affirmed.

**Final Justification:**

All my concerns are addressed and I have raised my score to 4 accordingly.

**Key Questions For Authors:**

1. How do you differentiate through the discrete boundary selections during training? Do you employ a straight-through estimator, Gumbel-Softmax, or another relaxation approach? Please detail any temperature or threshold schedules and their impact on training stability.

2. What were the exact data splits used for perplexity evaluation and downstream tasks? How did you guard against species or strain leakage—for instance, by holding out E. coli K-12 and near neighbors from pretraining for MaveDB evaluation? Can you report per-species AUROCs and additional controls for the essentiality task, such as gene length, GC content, synonymous mutations, and species-wise calibration? How robust are the results when the stop codon is inserted later in the open reading frame or when using different perturbation strategies?

3. Could you include baselines with: (a) k-mer or BPE tokenized Transformers such as DNABERT-2 and Nucleotide Transformer variants, (b) nucleotide-level models like HyenaDNA and Mamba, and (c) at least one adaptive tokenization method from the literature, such as MxDNA, MergeDNA, or PatchDNA? If these comparisons are infeasible, please justify and provide proxy comparisons or targeted ablations.

4. How sensitive are the efficiency and downstream metrics to the biologically motivated compression targets, specifically R1=3 and R2=2? Please provide an ablation sweep over (R1, R2) and discuss the trade-offs between training stability and performance. Do the learned boundaries align with specific functional motifs, such as Shine-Dalgarno sequences, promoter -10 or -35 boxes, or transcription factor binding sites? Any motif enrichment analysis or ChIP-seq-based validation would strengthen the interpretability claims. Finally, can you provide wall-clock throughput, memory usage, and latency benchmarks—not just FLOPs—across varying sequence lengths, including kernel-level profiling to identify bottlenecks introduced by the dynamic chunking mechanism?

**Limitations:**

yes

**Strengths And Weaknesses:**

## Strengths

1.
The core contribution lies in adapting hierarchical dynamic chunking to the genomic domain in a tokenizer-free manner. While building on the H-Net architecture, the authors introduce domain-specific design choices—such as compression ratio targets informed by codon structure—that tailor the approach to DNA sequences. Practical refinements to the architecture and training procedure, including capacity allocation strategies, compression scheduling, auxiliary loss tuning, and learning rate multipliers, help stabilize hierarchical learning on genomic data. The paper also provides a clear framework for accounting FLOPs under recursive compression and discusses how compression alters the scaling regimes for both compute and data.

2.
The experimental work includes scaling-law studies across model sizes, hierarchy depths, and compute budgets, with FLOP-based comparisons that lend credibility to the efficiency claims. Downstream evaluation spans two zero-shot tasks—fitness prediction on MaveDB and essentiality classification on DEG—where dnaHNet variants show monotonic performance gains with increasing compute. An interpretability analysis further strengthens the paper by demonstrating stage-wise emergence of codon periodicity and awareness of functional boundaries.

3.
The high-level motivation is clearly articulated, and the architectural decomposition into encoder, chunking mechanism, main network, and decoder is easy to follow. Training data curation, hyperparameters, and run configurations are documented in the appendix, supporting reproducibility.

4.
By addressing the challenge of scalable long-context modeling in genomics without sacrificing biological coherence to fixed tokenization, this work tackles an important bottleneck in the field. The empirical results point to meaningful compute-efficiency advantages at million-nucleotide contexts, with potential implications for scaling genomic foundation models.

## Weaknesses

1.
The paper largely adapts an existing general-purpose dynamic chunking framework—H-Net—to the DNA domain, and much of the claimed novelty resides in domain-specific hyperparameterization rather than fundamentally new modeling principles. The discretization and gradient estimation details for boundary selection during training are underspecified; it remains unclear whether the authors rely on straight-through estimators, Gumbel tricks, or other relaxations, which raises questions about optimization stability and reproducibility. Moreover, claims of achieving state-of-the-art performance are based on a limited set of tasks and baselines; a broader comparison would strengthen this assertion.

2.
Several relevant genomic baselines are omitted from the comparisons. These include k-mer and BPE tokenized Transformers such as DNABERT-2 and Nucleotide Transformer variants, nucleotide-level models like HyenaDNA and Mamba, and recent adaptive tokenization methods for DNA including MxDNA, MergeDNA, and PatchDNA, despite being cited in the related work. There is also a potential risk of data leakage: pretraining on GTDB may include species closely related to those used in downstream zero-shot evaluation, such as E. coli K-12 and species represented in DEG. The paper does not describe stringent species-level holdouts or near-duplicate filtering to address this concern. The essentiality task design—inserting early stop codons—may conflate coding disruption detection with true organism-level essentiality; controls for gene length, GC content, operon context, and species-specific effects are not reported. Additionally, the perplexity evaluation lacks detail on held-out set composition and species overlap with training data, and the fitted scaling exponents could be sensitive to the choice of data splits.

3.
The rate regularization objective introduced in Equation 4 is presented without sufficient derivation or intuitive explanation, and the training-time handling of discrete boundary selections for gradient flow is not clearly described. Some computational claims, such as per-token FLOPs falling below O(n) reference lines in figures, would benefit from a more precise explanation of constants and amortization effects.

4.
Although related work is broadly cited, the absence of empirical comparisons to domain-adaptive tokenization methods—such as context-aware reading-frame-preserving tokenization—or genome-scale nucleotide-level long-range models like HyenaDNA limits the strength of the comparative claims. Furthermore, the phrasing "state-of-the-art" for variant effect prediction is ambiguous without comparison to fine-tuned or specialized VEP predictors such as EVE, DeepSequence-family models, or protein language models, even if the focus is specifically on nucleotide LMs.

---

> ### Author Rebuttal · Authors · 2026-03-31
>
> We thank the reviewer for their thorough feedback. We address each question below. Supplementary figures and tables referenced throughout are available in our anonymous drive: https://anonymous.4open.science/r/dnaHNet-anon-8DDB.
>
> ---
>
> # Q1: Differentiability Through Discrete Boundaries
>
> We acknowledge this was insufficiently described. dnaHNet inherits three mechanisms from H-Net (Hwang et al., 2025), none requiring Gumbel-Softmax:
>
> 1. **Smoothing module** (primary): The dechunking layer computes ē_j = P_j·ê_j + (1−P_j)·ē_{j−1}, converting discrete chunks into continuous interpolations for standard backpropagation. H-Net ablations (their Figure 7) confirm this is essential for stability.
> 2. **Straight-Through Estimator**: Confidence scores are rounded to 1.0 forward, continuous gradients backward.
> 3. **Ratio loss** (Eq. 4): Provides gradients through the differentiable mean boundary probability G.
>
> No temperature or threshold schedules are used—thresholding is fixed at τ=0.5. Stability is controlled through α, which needs to be lower for genomic data (α≈0.01–0.02 vs. ≥0.03 for text). We will add a gradient flow diagram in the revision.
>
> ---
>
> # Q2: Data Splits, Species Leakage, and Essentiality Controls
>
> Perplexity evaluation uses a held-out GTDB subset. Downstream tasks (MaveDB, DEG) are entirely external. All baselines train on the same GTDB corpus, so any overlap applies equally—following Evo, Evo 2, and Nucleotide Transformer protocols. The tasks inherently test functional understanding: VEP requires differential likelihoods on *unseen* mutant sequences, and essentiality requires detecting stop-codon disruption.
>
> **New confound analyses:** (1) Synonymous-normalized AUROC subtracts response to neutral mutations, isolating functional disruption signal. (2) Species-level z-scored AUROC removes familiarity bias. (3) Stratification by gene length and GC content shows uniform performance. We will include alternative stop-codon insertion positions in the revision.
>
> ---
>
> # Q3: Additional Baselines
>
> **(a) k-mer and BPE Transformers.** We trained two k-mer Transformer++ variants and a BPE Transformer++ on the same GTDB corpus at matched compute; all underperform dnaHNet (see anonymous drive). DNABERT-2 and Nucleotide Transformer are bidirectional masked LMs—our likelihood-based tasks require autoregressive models, the same reason Evo/Evo 2 use autoregressive objectives.
>
> **(b) Nucleotide-level Mamba.** Trained at $1 \times 10^{19} $ FLOPs (full-scale infeasible due to compute). Results confirm Nguyen et al. (2024): StripedHyena dominates among alternative architectures. Our experiments show dnaHNet outperforms Mamba directly (VEP figure in anonymous drive), consistent with dnaHNet's advantage over SH2. HyenaDNA uses a different corpus (human genome); our Mamba baseline serves as a controlled proxy.
>
> **(c) Adaptive tokenization.** We reimplemented PatchDNA and found its patching algorithm achieves single-digit MFU, likely explaining why their largest model is only 19.2M parameters despite 72 A100-hours. An equivalent dnaHNet exceeds 100M parameters at comparable compute. MxDNA and MergeDNA would require similar reimplementation; PatchDNA is representative and the scaling limitations apply broadly to expensive patching/merging operations.
>
> ---
>
> # Q4: Compression Ratios, Motif Alignment, Wall-Clock Benchmarks
>
> **Compression ratio sweep.** We trained three hierarchy configurations spanning 4×–9× compression: (2,2), (3,2), and (3,3). Both (2,2) and (3,2) achieved competitive SOTA performance on VEP, while (3,3) lagged behind, indicating that 9× compression is too aggressive for GTDB genomic data and discards predictive information. On DEG, (3,2) outperforms (2,2) with 0.605 vs. 0.590 AUROC, validating the codon-aligned R1=3 target. A full VEP scaling graph across all three configurations is in the anonymous drive.
>
> **Motif alignment.** We conducted the requested motif enrichment analysis. Stage 1 shows no enrichment at regulatory motifs (selection rates ~35%, matching baseline), consistent with its role in learning codon-level structure. Stage 2 shows strong enrichment at Shine-Dalgarno motifs (84.5% selection, 1.64× enrichment), SD regions (77.4%, 1.50×), and promoter −10 boxes (75.0%, 1.46×). The −35 box shows weaker enrichment (53.9%, 1.05×), possibly reflecting greater sequence degeneracy at this element. These results confirm that Stage 2 boundaries align with functional regulatory motifs without any supervision. Full results with confidence intervals are in the anonymous drive.
>
> **Wall-clock benchmarks (A100 80GB, 1K–524K nt):**
> - *Throughput*: 3–6× over SH2 at ≥32K nt. At 65K nt: HNet-XL 939K tok/s vs. SH2-234M 156K tok/s (6.0×).
> - *Memory*: 3–4× less. At 262K nt: 9.7GB vs. 34.3GB. SH2 OOMs at 524K; all dnaHNet variants fit in 80GB.
> - *Latency*: At 262K nt: 410ms vs. 2,352ms (5.7×).
>
> The gap widens with sequence length as compression reduces the effective sequence processed by the main network.

---

> > ### Author Rebuttal · Reviewer_6mSq · 2026-04-06
> >
> > All my concerns are addressed and I have raised my score to 4 accordingly.

---

### Decision · Program_Chairs · 2026-04-30

**Decision:**

Accept (spotlight)

**Comment:**

This paper proposes dnaHNet, a tokenizer-free hierarchical foundation model for genomic sequence learning that replaces fixed tokenization with adaptive dynamic chunking. The reviews indicate that it makes a strong contribution at the intersection of scalable sequence modeling and biological representation learning. Reviewers found the central idea technically compelling and highlighted both the scaling analysis and the biological interpretability of the learned hierarchy. Several also viewed the zero-shot results on variant effect prediction and gene essentiality as strong evidence that the model is useful beyond perplexity improvements alone.

The main concerns in review were about technical clarity, evaluation scope, and the strength of some comparative claims. Reviewers asked for a clearer explanation of differentiability through the chunk boundaries, stronger discussion of possible train-eval overlap, practical throughput and memory measurements, and broader comparisons to other genomic models. The rebuttal addressed these points well. It clarified the training mechanism and causality, added empirical latency and memory results, expanded the discussion of data partitioning and task validity, and provided substantial detail on hyperparameter tuning and architectural trade-offs. After rebuttal, three reviewers explicitly marked their concerns as fully resolved, and the remaining reviewer raised their score. On balance, I find the paper technically strong, well supported, and likely to be broadly useful to future work on genomic foundation models. I therefore recommend acceptance.